# Robust AMBER Force Field Parameters for Glutathionylated Cysteines

**DOI:** 10.3390/ijms241915022

**Published:** 2023-10-09

**Authors:** Zineb Elftmaoui, Emmanuelle Bignon

**Affiliations:** UMR 7019 LPCT, Université de Lorraine and CNRS, F-54000 Nancy, France

**Keywords:** S-glutathionylation, post-translational modifications, AMBER force field parameters, redox modifications, molecular dynamics simulations

## Abstract

S-glutathionylation is an oxidative post-translational modification, which is involved in the regulation of many cell signaling pathways. Increasing amounts of studies show that it is crucial in cell homeostasis and deregulated in several pathologies. However, the effect of S-glutathionylation on proteins’ structure and activity is poorly understood, and a drastic lack of structural information at the atomic scale remains. Studies based on the use of molecular dynamics simulations, which can provide important information about modification-induced modulation of proteins’ structure and function, are also sparse, and there is no benchmarked force field parameters for this modified cysteine. In this contribution, we provide robust AMBER parameters for S-glutathionylation, which we tested extensively against experimental data through a total of 33 μs molecular dynamics simulations. We show that our parameter set efficiently describes the global and local structural properties of S-glutathionylated proteins. These data provide the community with an important tool to foster new investigations into the effect of S-glutathionylation on protein dynamics and function, in a common effort to unravel the structural mechanisms underlying its critical role in cellular processes.

## 1. Introduction

Among the post-translation modifications (PTMs) regulating cellular processes, protein S-glutathionylation (SSG) has a central role in redox signaling pathways, yet remains largely understudied. This PTM consists of the addition of the tripeptide glutathione (GSH) (a very abundant low-molecular weight thiol) to target cysteines and acts as a key player in the redox regulation of protein function in humans, plants, and bacteria [1]—see Figure 1.

SSG participates in cellular homeostasis and has a crucial redox regulatory role in a large panel of cellular processes (e.g., apoptosis, proliferation, DNA compaction [2,3,4,5]). In response to increased Reactive Oxygen Species (ROS) amounts in the cell, SSG levels are upregulated to modulate the activity of a plethora of key proteins, from signaling proteins to transcription factors and ion channels [6,7]. For instance, SSG participates in 1-cys peroxiredoxin activation [8], influences endothelium-dependent vasodilation by decreasing endothelial nitric oxide synthase activity [9], and regulates the elasticity of cardiomyocytes by altering the folding of the giant elastic protein titin [10]. It is also thought to have a protective role against irreversible oxidation of cysteine (e.g., sulfonylation), which could be generated under oxidative stress, and plays a crucial role in the regulation of proteins involved in immune/inflammatory response to infection [11,12]. Elevated GSH and SSG levels have been observed in numerous diseases associated with oxidative stress and are most probably linked to anticancer drug resistance [13,14,15,16], further underlining the high biological relevance of this PTM in health and disease.

In cells, S-glutathionylation can be formed non-enzymatically [11]: by S-thiolation involving GSH and an oxidized cysteine, by S-thiolation of an oxidized GSH with a cysteine, or by thiol/disulfide exchange between GSH and a cysteine thiolate group [15]. Although this modification was thought to occur in highly oxidative conditions, it is now well known that it happens in physiological conditions and that it is an important player in physiological redox signaling and cellular homeostasis. SSG levels are also regulated by dedicated enzymes, which act as writers and erasers of S-glutathionylated cysteines (CSGs). Recent advances in experimental techniques dedicated to the study of this kind of reversible redox modifications brought out important information about SSG formation and removal [17], but many aspects remain to be uncovered regarding this topic. Glutathione transferase enzymes (GSTs) have been shown to catalyze protein S-glutathionylation, and the glutaredoxin Grx1 is the main one responsible for protein deglutathionylation in mammalian cells, with higher activity than the thioredoxin system [11].

**Figure 1 ijms-24-15022-f001:**
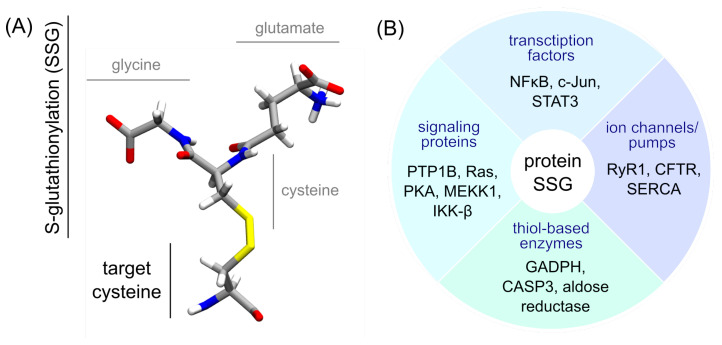
(**A**) Structure of a glutathionylated cysteine. The glycine, cysteine, and glutamate residues within the glutathionylation moiety (SSG) are labeled in grey. The target cysteine backbone atoms are displayed as if embedded in a protein (without capping). (**B**) A few examples of proteins regulated by S-glutathionylation, with respect to their function [7,18,19].

Despite a growing interest by scientists in unraveling the regulatory mechanisms of protein glutathionylation, the amount of all-atom molecular dynamics studies present in the literature remains scarce compared to other mainstream PTMs such as phosphorylation [20]. In most recent computational studies, Hyslop et al. reported the structural impact of SSG on the GADPH protein’s structure [21]; Moffet et al. described the SSG-induced allosteric effects in a plant kinase [22]; Zhou et al. investigated the effect of SSG on ATP/ADP binding to the adenine nucleotide translocator 2 protein [23]; Zaffagnini et al. provided clues about the SSG-triggered collapse of a soluble tetrameric plant protein [24]; Hameed studied the inhibition of the triose-phosphate isomerase by SSG [25]. However, CSG parameterization is rarely detailed, sometimes not even mentioned, and parameter sets are not provided. Nevertheless, a thorough validation of CSG parameters against experimental data needs to be performed and made available to the community in order to ensure the reproducibility and the relevance of the simulations of CSG-harboring systems.

In order to palliate the absence of any extensively tested force field parameters for S-glutathionylated cysteines, we report here a set of AMBER parameters and their validation using eleven available experimental structures of S-glutathionylated proteins. The parameters were tested owing to microsecond-range MD simulations, performed starting either directly from the S-glutathionylated experimental structure or from an Alphafold-predicted unmodified structure that was further S-glutathionylated in silico. Indeed, most of the future computational investigations of S-glutathionylated proteins will probably face the unavailability of any experimental starting structure for their simulations, hence the importance of testing the parameter set in similar conditions. We show that our AMBER-type parameter set succeeds at reproducing the experimental global and local structural properties of S-glutathionylated proteins, both starting from experimental and in silico-mutated structures. When starting from an in silico-generated model of the modified protein, we advise to use multiple starting positions featuring different χ side chain angles for the S-glutathionylated cysteine. This work provides the community with a robust tool to foster new initiatives aiming at unveiling the role of protein S-glutathionylation in cells.

## 2. Results

### 2.1. Experimental Structure Curation

The reference structures of experimental S-glutathionylated proteins were curated from the Protein Data Bank by imposing a query on the GSH residue. Among the structures found, only those without any ligand, cofactor, other PTM, metal center, transmembrane domain, or internal missing region were kept. Structures with a resolution higher than 2.0 Å were evicted. This resulted in a selection of eleven proteins, mostly involved in redox processes such as thioredoxins, glutaredoxins, and glutathione transferases, besides the structure of the human lysosyme. All of these structures were obtained by X-ray crystallography, and three of them were homodimers harboring two S-glutathionylation sites (TvGSTO2C, hGSTO2, and PtGSTF1). These structures are listed in Table 1—see the Appendix A for more details.

It is noteworthy that none of these proteins’ activity is regulated by S-glutathionylation: this modification is a production of the deglutathionylation activity of ScTrx1, PtGrxS12, AtGrxC5, ScGrx2, PtGSTL1 and 3, hGSTO2, and TxGSTO2C, an intermediate in the recycling mechanisms of PtTrxL2.1 catalytic cysteine, a mimic of the catalytic site of PtGSTF1 for the crystallization, and a mimic of a disulfide bridge intermediate for the acquisition of the hLyso crystal structure. For each of these systems, an unmodified model was generated in silico using the Alphafold-based Colabfold online tool. It is noteworthy that the PDB file of the dimeric TvGSTO2C crystal structure (PDB ID 6SR8) displays a peculiar arrangement of the two monomers, different from the regular glutathione transferase quaternary structure—see Appendix A. Yet, as the model generated by Colabfold appears to exhibit a regular glutathione transferase fold, the system was simulated as such, and analyses were run on separated monomers only. Besides, it is also important to underline that, in the experimental structure of PtTrxL2.1, some atoms of CSG were missing (the terminal NH${}_{3}^{+}$ amino group of the side chain), presuming a high flexibility in this case—see Appendix A.

### 2.2. Global Structural Features of the Systems

The global structural features of each system were calculated from the 3 × 1 μs MD ensembles, in order to probe if our CSG parameter combined to the ff14SB set would robustly describe the proteins 3D organization. For the sake of clarity, we further refer to “crys” and “AF” simulations for the MD simulations from the crystal or from the Colabfold structures, respectively.

The first structural descriptor to scrutinize was the RMSD of each MD ensemble with respect to the reference experimental structure. We observed very good trends with average deviations systematically lower than 2.4 Å when considering all the atoms and lower than 1.6 Å for the backbone atoms—see Figure 2 and Appendix A.

Interestingly, the MD ensembles generated from the in silico models (AF simulations) exhibited average RMSD values within the range of the ones obtained for MD simulations starting from the experimental structure (crys simulations), both considering all the atoms or only the backbone atoms. Of note, the RMSD values of the dimers were 2.36 ± 0.22 Å and 2.22 ± 0.28 Å for AF and crys PtGSTF1 and 2.79 ± 0.33 Å and 2.20 ± 0.23 Å for AF and crys hGSTO2 when considering all the atoms in the calculation—see Appendix A. The highest standard deviations were observed for the hLyso (up to 0.33 Å for all atoms), one PtGSTF1 monomer (up to 0.52 Å for backbone atoms), and one TvGSTO2C monomer (up to 0.43 Å for all atoms), which are mostly caused by the fluctuations of some loops, as evidenced from superimposing the representative structures of the major conformations identified by the cluster analysis—see Figure 3. Of note, the experimental resolutions were in the range of 1.2–1.95 Å, excluding hydrogen atoms, which underlines the robustness of our parameters combined with ff14SB in simulating the conformational behavior of these S-glutathionylated proteins.

Another global descriptor of proteins is their radius of gyration, which provides information about their compactness. The average values calculated on the MD ensemble were in very good agreement with the reference values from the crystal structures, with a maximum deviation of only 0.4 Å observed for ScTrx1 AF simulations—see Table 2. The values obtained from the AF simulations were again within the standard deviation of the ones calculated from the crys simulations, highlighting the robustness of our protocol. It is noteworthy the values for TvGTO2C are not given considering the different quaternary structures of the AF and crys systems, as mentioned above.

### 2.3. CSG Structural Features

In order to have a thorough validation of our CSG parameters, an in-depth investigation of the modified cysteine sites’ structural features and their interaction networks was carried out.

The CSG RMSD trends varied much more than what was observed for the protein RMSD—see Figure 4. While for most of the systems, there was a very good agreement with the experimental reference, illustrated by average RMSD values lower than 1.5 Å, some cases stood out with values up to 2.38 ± 0.28 Å for hLyso AF simulations. The PtTRXL2.1 and ScTrx1 systems also exhibited higher RMSD values than the others for both AF and crys simulations, which can be rationalized by a weak interaction network around the CSG residue, which resulted in its great flexibility, as described below.

It is noteworthy that the CSG RMSD time series of the AF simulations showed that, even when starting from an in silico-generated system, the CSG parameterization allowed retrieving the CSG reference geometry, characterized by a drop in the RMSD after some time of simulation—see Figure 5 and Appendix A. We can cite as striking examples the cases of PtGSTL1 and ScGRX2, for which the RMSD values of the CSG were up to 2.5–3.0 Å at the beginning of the AF simulations and, then, dropped below 1.0 Å. For AtGrxC5, PtGSTL3, and PtGrxS12, the RMSD of the CSG at t = 0 ns was around 2.0 Å and dropped rapidly around 1.0 Å. In the case of the PtGSTF1 dimer, exchanges between two conformations close to the reference were observed, characterized by RMSD values of around 1.0 Å for the first one and 1.5 Å for the second one. The second conformation was also sampled in the crys simulation, yet to a lesser extent—see Appendix A. This can be explained by small changes in the interaction network of the CSG site (especially sampled in Monomer 2), which is described below. Besides, the CSG RMSD values in hGSTO2 Monomer 2 of the first AF simulation replicate (MD1) did not show any drop, contrary to the other replicates (MD2 and MD3), explaining the slightly higher average value (1.58 ± 0.28 Å) compared to the crys simulations (1.24 ± 0.24 Å). This was also the case for the CSG site in the hGSTO2 monomer 1, whose RMSD dropped from 2.0 to 1.5 Å after only 500 ns and 800 ns of simulation in replicates MD1 and MD2, respectively, therefore exhibiting an average RMSD of 1.58 ± 0.28 Å in AF simulations compared to 1.24 ± 0.24 Å in crys simulations.

An important structural feature to check was the dihedral angle defining the new bond formed upon the addition of the glutathione onto the cysteine residue, i.e., the dihedral angle around the S-S bond. The values from the AF and crys simulations were in excellent agreement with the reference structure, i.e., mainly +90${}^{\circ }$—see Appendix A. PtGSTF1 is the only system where the CSG S-S angle was found at −90${}^{\circ }$ (only in Monomer 1), which was accurately reproduced in both AF and crys simulations. We also note that the values of this angle were rather centered around the average value, except for PtTrxL2.1, in which the distribution was broader for both AF and crys simulations, but remained very much centered around the reference value of +90${}^{\circ }$. This might be due to the lack of strong hydrogen bonds involving the CSG side chain in this system (see the details below), also resulting in higher CSG RMSD values in our simulations and in high B-factor values in the experimental structure (38.9 ± 7.6 Å${}^{2}$ on average for the CSG atoms).

Concerning systems with the highest CSG RMSD deviations (i.e., hLyso, ScTrx1, and PtTrxL2.1), the superimposition of the major CSG conformations sampled suggests that the orientation of the CSG side chain in the AF simulations was reversed with respect to the reference structure—i.e., the glutamate and the glycine sides were exchanged; see Figure 6A. However, this does not explain why a high average of the CSG RMSD was also observed in the crys simulations, which might be caused by a weak interaction network around the CSG, as described below. The RMSD time series showed a conformational drift in PtTrxL2.1 crys simulations with an increase of the RMSD from 1.7 Å to 2.3 Å in replicates MD1 and MD3.

A similar behavior was observed in the hLyso crys simulations with exchanges between several states in all replicates. Finally, the CSG RMSD values in the ScTrx1 AF and crys simulations were found to fluctuate greatly (2.12 ± 0.30 Å and 2.14 ± 0.33 Å, respectively), without showing any stable conformation. An in-depth look at the interaction network involving the CSG residues helped to rationalize these observations.

### 2.4. CSG Interaction Network

Native interactions with the CSG side chain are listed for each system (see Appendix A) and monitored along the AF and crys simulations. The CSG moieties involved in hydrogen bonds with the surroundings are the atoms of the amide moieties that result from dehydration steps of GSH formation (O2, OE1, H4, and H2), the carboxylate of the glycine end (C3), and the carboxylate and amino groups of the glutamate end (C1 and N3)—see Figure 7A. Interestingly, conserved interaction patterns can be pinpointed in glutaredoxins, which involve a valine and a serine or threonine backbone atoms (V74 and T88 in both AtGrxC5 and PtGrxS12 and V75 and S89 for ScGrx2). Similar patterns were also found in the glutathione transferase proteins, with interactions between CSG.O2 and valine backbone atoms (PtGSTL1 V79, PtGSTL3 V84, hGSTF1 V56, TvGSTO2C V56), between CSG.C1 and a serine (PtGSTL1 S92, PtGSTL3 S97, hGSTF1 S69, TvGSTO2C S81), and between CSG.N1 and a glutamate (PtGSTL1 E91, PtGSTL3 VE96, hGSTF1 E68, TvGSTO2C E80). hGSTO2 exhibited homologous interactions for CSG.O2 and CSG.C1 with I72 and S96, respectively. Consistent with the CSG RMSD trends, the simulations results were in excellent agreement with the experimental reference. Most of the native interactions were observed in our simulations, and the AF models generally succeeded in retrieving the hydrogen bond (HB) network identified in the experimental structures—see Appendix A.

All native hydrogen bonds were very efficiently reproduced in AtGrxC5, PtGSTL1, PtGSTL3, PtGrxS12, ScGrx2, and TvGTO2C, for both the crys and AF simulations. The distance values in the AF simulations of hGSTO2 and PtGSTF1 showed broad distributions, probably because of the long simulation time necessary to retrieve the native CSG orientation—see Appendix A. As described previously in the CSG RMSD plot analysis, in some MD replicates, the CSG geometry evolved to match the reference only in the late moments of the simulation. This was the case for hGSTO2 Monomer 1, and in one of the replicates (MD1), the CSG conformation in Monomer 2 did not match exactly the native orientation. Indeed, in this replicate, it failed at retrieving the experimental HB network, especially the interaction between CSG.O2 and I72.H in both monomers. In Monomer 1 only, the CSG.C1—S96.HG hydrogen bond was not observed, yet CSG.C1 still strongly interacted with S96.H, suggesting that this is the major interaction for this site (also observed in the other glutathione transferase systems). The same trends were found for PtGSTF1, for which the CSG can adopt the same orientation as in the experimental structure, but the HB network can deviate. The interactions for Monomer 2 tended to exhibit distributions with peaks matching the reference, but this was less pronounced for Monomer 1. Yet, the systematic presence of matching peaks suggests that these conformations were actually sampled, and the system might need more simulation time to stabilize itself. While the AF simulations might require more extended sampling, the results for the crys simulations showed in any case very robust trends, in agreement with the reference.

The hLyso is a peculiar case, because it showed good results for the crys simulations, while the AF simulations failed to correctly reproduce the native HB network—see Figure 7B. In fact, the starting structure generated in silico features of a CSG side chain with an orientation opposite to what is found in the reference structure. During the simulation, the CSG was trapped in this conformation and could not rotate to retrieve the native interactions that were found to be stable in the crys simulations, especially for CSG.O2 and H2 backbone atoms. We suggest that starting from multiple structures with opposite orientations of the CSG side chain or resorting to enhanced sampling methods would be good solutions to overcome this problem.

It is noteworthy that the native interactions in PtTrxL2.1 and ScTrx1 were found to be rather unstable in our simulations, with conformational fluctuations of the CSG side chain, as suggested by the CSG RMSD values—see Figure 7C,D. As a matter of fact, the only native interactions found for these two systems were a couple of HBs involving the OE1, O2, or H2 atoms at the center of CSG side chain. No strong HB involving the CSG charged termini could be observed, probably because the CSG is highly solvent-exposed in these two systems.

## 3. Discussion

Force field parameters for S-glutathionylated cysteines were generated and extensively tested against data from eleven curated experimental proteins bearing S-glutathionylation. Molecular dynamics simulations of two types of starting systems were conducted: (i) directly from the crystal structure or (ii) from a 3D geometry predicted by the Alphafold-based Colabfold facility and S-glutathionylated in silico using the leap module of AMBER20. As only a few experimental structures of S-glutathionylated proteins are available on the PDB, it was important to test our parameters on a fully in silico starting structure, which might reflect the most-probable situation encountered in future studies involving this modification.

The CSG parameter set was generated by calculating the RESP charges and assigning atom types from ff14SB. All the needed parameters to describe the CSG were already available in the ff14SB force field, and no specific parameterization was required. The CSG atom names were assigned to match the ones of the GSH entry on the Protein Data Bank, in order to facilitate their use for other systems. The AMBER library files are available online (https://github.com/emmanuellebignon/CSG-parameters, accessed on 19 September 2023).

An extensive validation of the CSG parameters was conducted. Eleven reference structures were identified in the PDB (see Table 1, which mostly featured redox-related proteins such as glutaredoxins and glutathione transferases. Three replicates of 1 μs were carried out from the two types of starting systems, for each of the curated protein. The simulations starting from the crystal structure (crys simulations) generated conformations in excellent agreement with the reference data, both in terms of global (protein) and local (CSG site) features. The protein RMSD was systematically found below 2.3 Å when considering all the atoms (including hydrogens) and below 1.5 Å when taking into account only the backbone atoms—see Figure 2. The overall performances of our parameters were very good considering the resolution of the chosen reference structures, ranging from 1.20 to 1.95 Å. Importantly, the conformations of the dimeric systems bearing two CSG sites were also very well reproduced in the crys simulations. The CSG features were also efficiently reproduced in the crys simulations, with RMSD values below 1.3 Å for most of the systems. Higher deviations were observed for the PtTrxL2.1 (1.96 ± 0.37 Å) and ScTrx1 (2.14 ± 0.33 Å) systems, mostly due to the weak interaction network around the CSG residue, resulting in a high flexibility of its side chain. The hLyso also exhibited slightly higher RMSD values for the CSG (1.61 ± 0.52 Å), which can be rationalized by the conformational exchanges in the simulations due to the movements of the disordered loop facing the CSG residue.

Simulations from the in silico starting structures provided more contrasted results. If the results showed very good trends for most of the systems, with global and local features rapidly converging towards the reference values, some cases showed that such a scenario must be taken with care. The most-striking example illustrating this is the hLyso case, for which the AF simulations failed to reproduced the CSG local features of the experimental structure—see Figure 6 and Figure 7. This was due to the fact that the initial orientation of the CSG side chain was reversed with respect to the reference one. While in other systems. the CSG reference geometry and interaction network were retrieved after some sampling time (see Appendix A), for hLyso, the side chain was trapped in the initial wrong conformation in all replicates. We suggest that, when starting from an in silico structure, multiple replicates should be launched from geometries featuring different CSG side chain orientations or, eventually, we suggest using enhanced sampling methods in order to boost the sampling of different conformations for the CSG residue. Given the excellent results obtained for the crys simulations, one should manage to converge to similar results in AF simulations by overcoming this starting orientation/sampling problem.

## 4. Materials and Methods

All MD simulations, QM calculations, and structural analyses were performed with NAMD3 [36], Gaussian16 [37], and the Ambertools20 [38], respectively.

### 4.1. Generation of the CSG Parameters

The initial geometry of an S-glutathionylated cysteine was taken from the CSG entry on the Protein Data Bank website and capped with an acetyl (-OCH${}_{3}$) and a methylamino (-NHCH${}_{3}$) group on the N-term and C-term ends, respectively. This structure was optimized at the B3LYP/6-31+G* level, using a Polarizable Continuum Model to model an implicit solvent (water, ϵ = 80) and setting the charge to −1 and the spin multiplicity to 1. The frequency calculation was performed in order to ensure the convergence of the minimization to a local potential energy well, characterized by the absence of negative vibrational frequency. The optimized geometry was extracted and subjected to a charge calculation using the Merz–Singh–Kollman scheme [39], at the HF/6-31+G* level.

The antechamber AMBER20 module was then used to extract the geometry, fit RESP charges, and assign atom types into a mol2 file. Atom names were kept as in the CSG entry of the PDB as much as possible. The charge of the capping atoms was set to 0; the backbone atom charges were set to match the cysteine charges from ff14SB [40]; the global charge difference was equally distributed on the side chain atoms (Δ of −0.0018 elementary charge per atom) to ensure a total residue charge of −1. The atom types were corrected to match these of ff14SB for the backbone atoms. For the side chain, the atom types assigned by antechamber were kept after checking these were correct. The parmchk2 AMBER20 module was used to check the presence of missing parameters to describe the CSG residue, but none were identified. An AMBER library file for the CSG residue was finally created from the mol2 file and the ff14SB parameters using the tleap AMBER20 module, while removing the capping atoms and setting the amino acid connectivity. The parameter files are available on our Github: https://github.com/emmanuellebignon/CSG-parameters (accessed on 19 September 2023).

### 4.2. System Setup

Each crystal structure was downloaded from the PDB and submitted to the propka 3 software [41] for protonation state assignment. Crystallographic ions and waters were removed, and C- and N-terminal domains were reconstructed when needed—see the details in Appendix A. Besides, the structures of the unmodified systems were generated using Alphafold2 through the Colabfold online facility [42,43], using the same primary sequence as the corresponding crystal structure. These initially unmodified structures were mutated in silico with tleapto retrieve the S-glutathionylated sites, in order to probe how robustly we could predict the experimental structures from a fully in silico model, which will mostly be the actual situation in further studies of S-glutathionylated proteins. Protonation states were assigned with respect to the propka 3 results for the experimental structure.

Each system was soaked into a TIP3P [44] water box (truncated octahedron) of 12 Å buffer, neutralized with Na${}^{+}$ and Cl${}^{-}$ counter-ions [45]. The topology and coordinates parameters were generated using tleap, using the ff14SB AMBER force field [40] and the CSG library designed in-house.

### 4.3. Molecular Dynamics Simulations Protocol

Each system was first minimized in 15,000 steps using the conjugate gradient method. The system was then thermalized to 300K and equilibrated in NPT along three runs of 600 ps with decreasing constraints on the solute. The time step was then changed from 2 fs to 4 fs by applying the Hydrogen Mass Repartitioning scheme [46], in the 1 μs production runs. Three replicates were performed for each system, and the information and checkpoint files were output every 40 ps. The Particle Mesh Ewald approach [47] was applied for electrostatic calculations, with a 9 Å cutoff. The simulations were carried out in NPT, with a Langevin thermostat (1 ps${}^{-1}$ collision frequency) and a Langevin piston barostat.

MD trajectories were visualized and centered using the VMD 1.9.3 software [48]. The RMSD, radii of gyration, distances, and dihedral descriptor calculations, as well as the clustering analysis were performed using the cpptraj AMBER20 module. Statistical values were computed and plotted with the ggplot2 R package in Rstudio 2023.06.2 Build 561 [49,50].

## 5. Conclusions

In conclusion, the combination of our CSG parameters with the ff14SB AMBER force field succeeded in reproducing the experimental global and local features of the experimental S-glutathionylated references. Simulations from the crystal structures provided conformations in excellent agreement with the reference data. However, special care must be given when starting from in silico-generated structures, as the CSG side chain can be trapped in an initial wrong conformation. We advise multiplying the MD replicates with different CSG orientations, in order to efficiently probe the conformational behavior. The use of enhanced sampling methods should also allow overcoming this issue. This AMBER parameter set for the CSG is a robust tool for future computational investigations of S-glutathionylated systems, which will be of great importance towards the understanding of the S-glutathionylation mechanisms of action as a redox post-translational regulator and its link to disease onset and progression.

## Figures and Tables

**Figure 2 ijms-24-15022-f002:**
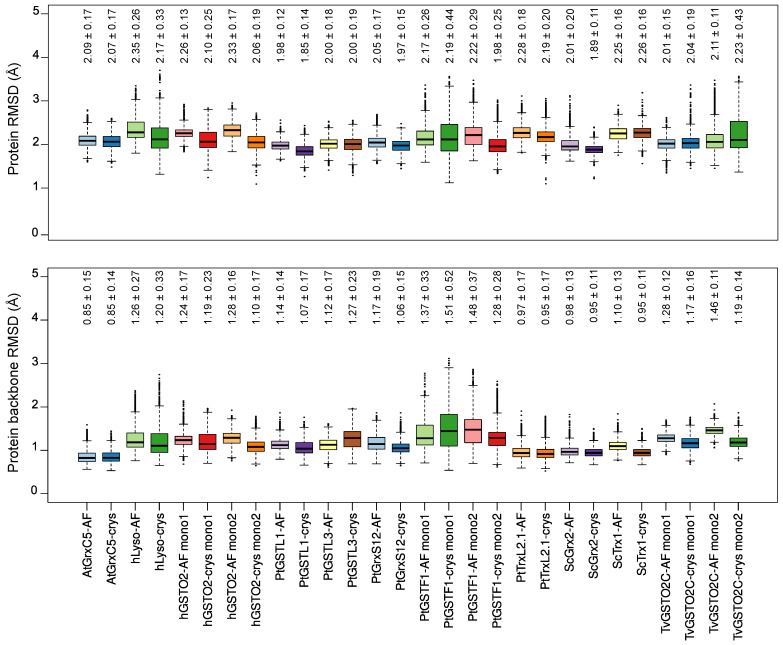
Boxplots of the RMSD values in Å of each system, considering all atoms (**top**) or the residues’ backbone atoms (**bottom**). RMSD values for dimers are shown by monomer (Mono 1 or Mono 2). The black lines inside each boxplot display the median values. The average and standard deviation values are written above each box.

**Figure 3 ijms-24-15022-f003:**
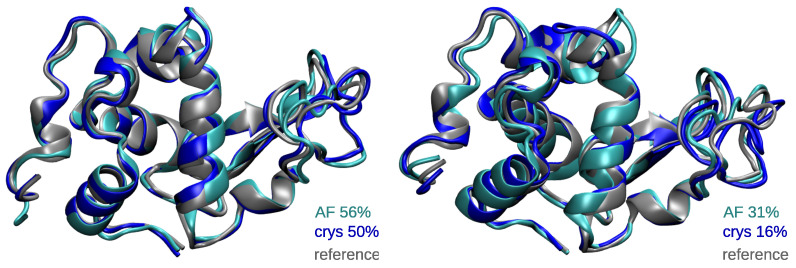
Representative structures of the two main conformations of hLyso as calculated from the cluster analysis of MD ensembles. The structures of the crystal reference and from the AF and crys simulations are displayed in gray, cyan, and blue, respectively. The percentage of occurrence of each cluster (primary on the left and secondary on the right) is also given.

**Figure 4 ijms-24-15022-f004:**
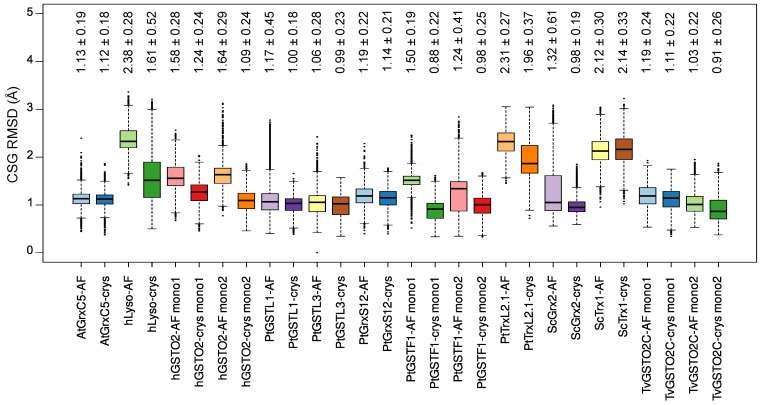
Boxplots of the S-glutathionylated cysteine (CSG) RMSD values in Å for each system. RMSD values for dimers are shown by monomer (Mono 1 or Mono 2). The black lines inside each boxplot display the median values. Average and standard deviation values are written above each box.

**Figure 5 ijms-24-15022-f005:**
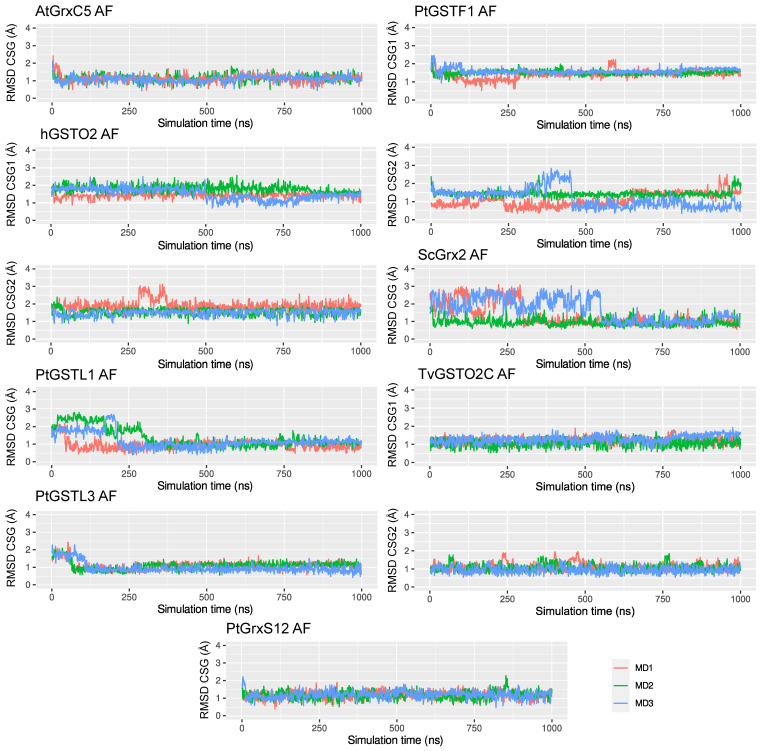
Times series of CSG RMSD values in Å of the three replicates for AF simulations of AtGrxC5, hGSTO2, PtGSTL1, PtGSTL3, PtGrx12, PtGSTF1, ScGrx2, and TvGSTO2C, which show better trends. Plots for the dimeric systems (hGSTO2, PtGSTF1, and TvGSTO2C) are paired in the columns, with CSG1 and CSG2 referring to Monomers 1 and 2, respectively.

**Figure 6 ijms-24-15022-f006:**
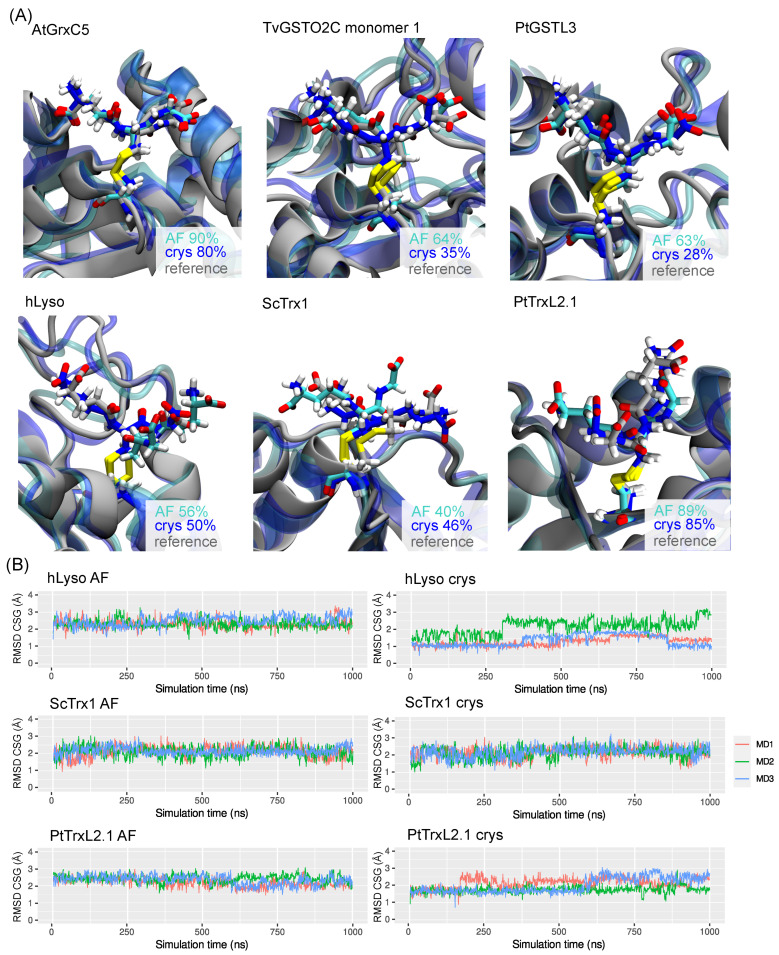
(**A**) Representative structures of the major conformations of the CSG in AtGrxC5, TvGSTO2C Monomer 1, PtGSTL3, hLyso, ScTrx1, and PtTrxL2.1 as calculated from the cluster analysis of MD ensembles. Structures of the crystal reference and from the AF and crys simulations are displayed in gray, cyan, and blue, respectively. The percentage of occurrence of the clusters is also given. (**B**) Times series of the CSG RMSD values in Å of the three replicates for the AF (left) and crys (right) simulations of hLyso, ScTrx1, and PtTrxL2.1.

**Figure 7 ijms-24-15022-f007:**
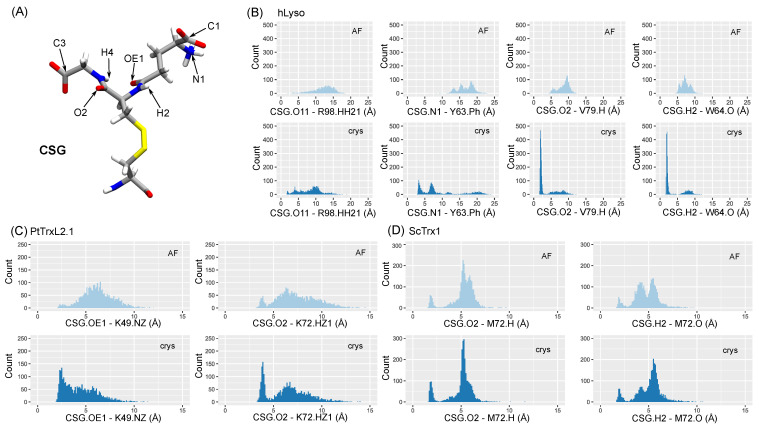
(**A**) CSG structure, highlighting the side chain atoms used to monitor hydrogen bonds. (**B**–**D**) Distribution of distances corresponding to the native interactions identified in the reference structures of hLyso, PtTrxL2.1, and ScTrx1, respectively. Both distributions for AF (cyan) and crys (blue) simulations are shown. Cation–π interactions involving Y63 in hLyso are monitored as distances between CSG.N1 and the center of mass of the aromatic ring heavy atoms of Y63 (Y63.Ph).

**Table 1 ijms-24-15022-t001:** List of the curated experimental reference structures providing the protein name, the organism, the abbreviation used for the system in this study, the PDB ID of the structure, and the S-glutathionylation site.

Protein Name	Organism	Abbreviation	PDB ID	Resolution (Å)	Oxidation Site
Thioredoxin 1	*Saccharomyces cerevisiae*	ScTrx1	3F3R [26]	1.80	C30
Thioredoxin-like2.1	*Populus tremula x tremuloides*	PtTrxL2.1	5NYN [27]	1.60	C67
Glutaredoxin S12	*Populus tremula x tremuloides*	PtGrxS12	3FZ9 [28]	1.70	C29
Glutaredoxin C5	*Aradobidopsis thaliana*	AtGrxC5	3RHB [29]	1.20	C29
Glutaredoxin 2	*Saccharomyces cerevisiaes*	ScGrx2	3D5J [30]	1.91	C30
Glutathione transferase λ1	*Populus trichocarpa*	PtGSTL1	4PQH [31]	1.40	C36
Glutathione transferase λ3	*Populus trichocarpa*	PtGSTL3	4PQI [31]	1.95	C41
Glutathione transferase ω2	*Homo sapiens*	hGSTO2	3Q19 ^1^ [32]	1.90	C29
Glutathione transferase ω2C	*Trametes versicolor*	TvGSTO2C	6SR8 ^1^ [33]	1.94	C30
Glutathione transferase F1	*Populus tremula x tremuloides*	PtGSTF1	4RI7 ^1^ [34]	1.80	C30
Lysosyme	*Homo sapiens*	hLyso	1HNL [35]	1.40	C95

^1^ Homodimeric structures, harboring two symmetric oxidation sites.

**Table 2 ijms-24-15022-t002:** Average and standard deviation of the radius of gyration (Rg) for simulations from the crystal structure (crys) and from the Colabfold structure (AF). The reference value calculated from the experimental structure is also given. All values are in Angströms.

System	Reference Rg	AF Simulations Rg	crys Simulations Rg
AtGrxC5	13.2	13.3 ± 0.1	13.3 ± 0.1
hLyso	14.3	14.4 ± 0.1	14.4 ± 0.1
hGSTO2	22.6	22.4 ± 0.2	22.7 ± 0.2
PtGSTL1	16.9	17.2 ± 0.1	17.2 ± 0.1
PtGSTL3	17.1	17.4 ± 0.4	17.4 ± 0.1
PtGrxS12	13.2	13.4 ± 0.1	13.4 ± 0.1
PtGSTF1	20.7	21.0 ± 0.1	21.0 ± 0.1
PtTrxL2.1	13.5	13.7 ± 0.1	13.7 ± 0.1
ScGrx2	13.0	13.4 ± 0.1	13.4 ± 0.1
ScTrx1	12.4	12.8 ± 0.1	12.7 ± 0.1

## Data Availability

All MD input files and post-processing scripts are available on Github. https://github.com/emmanuellebignon/CSG-parameters (accessed on 19 September 2023).

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
