# Peer review of "Robust AMBER Force Field Parameters for Glutathionylated Cysteines"

_ijms, 2023, doi:10.3390/ijms241915022_

Round 1
Reviewer 1 Report
The main purpose of this article is to provide a reliable set of AMBER force field parameters for simulating proteins containing S-glutathionylated cysteine (CSG). CSG is a post-translational oxidative modification that participates in the regulation of many cellular signaling pathways, but its effects on protein structure and activity are not well understood, and there is a lack of atomic-level structural information. The authors used quantum chemistry calculations to obtain the geometric structure and charge distribution of CSG, and combined them with the popular ff14SB force field to perform molecular dynamics (MD) simulations, and demonstrated that the newly developed force field parameter set can effectively describe the structural properties of CSG-containing proteins. The reviewer thinks that this work provide an useful tool for the community, which can facilitate the study of the impact of CSG on protein dynamics and function. The article is well written and logically organized, and the obtained CSG force field parameters are validated from multiple perspectives. It is worthy of being recommended as acceptable.
Some comments:
1. The authors used the newly developed CSG force field parameters to simulate the structural changes of various proteins before and after being modified by CSG. Can they discuss and compare them with some relevant experimental data for a specific protein?
2. What is the molecular mechanism of the functional changes of the protein after being modified by CSG? For example, how does the binding free energy change? If the authors can provide some related data, that would be the best; or they can analyze or prospect this in the discussion section.
Author Response
The main purpose of this article is to provide a reliable set of AMBER force field parameters for simulating proteins containing S-glutathionylated cysteine (CSG). CSG is a post-translational oxidative modification that participates in the regulation of many cellular signaling pathways, but its effects on protein structure and activity are not well understood, and there is a lack of atomic-level structural information. The authors used quantum chemistry calculations to obtain the geometric structure and charge distribution of CSG, and combined them with the popular ff14SB force field to perform molecular dynamics (MD) simulations, and demonstrated that the newly developed force field parameter set can effectively describe the structural properties of CSG-containing proteins. The reviewer thinks that this work provide an useful tool for the community, which can facilitate the study of the impact of CSG on protein dynamics and function. The article is well written and logically organized, and the obtained CSG force field parameters are validated from multiple perspectives. It is worthy of being recommended as acceptable.
We thank the reviewer for carefully reading our manuscript and for his/her comments, which are addressed below.
Some comments:
- The authors used the newly developed CSG force field parameters to simulate the structural changes of various proteins before and after being modified by CSG. Can they discuss and compare them with some relevant experimental data for a specific protein?
We thank the reviewer for underlining this point. We focused on the structural deviations of our model with respect to the experimental 3D structure, in order to quantify how performant was our force field parameters. Unfortunately, among the eleven proteins curated as reference experimental structures, none exhibit an activity that is regulated by S-glutathionylation. Instead, the S-glutathionylation states are mainly intermediates of (de)glutathionylation processes, or mimic of disulfide bridges used for the crystals acquisition. We described more into details the purpose of S-glutathionylation on the eleven proteins used for the benchmark (in section 3.1. Experimental structure curation) in the new version of the manuscript, in order to clarify this point.
- What is the molecular mechanism of the functional changes of the protein after being modified by CSG? For example, how does the binding free energy change? If the authors can provide some related data, that would be the best; or they can analyze or prospect this in the discussion section.
We thank the reviewer for this useful comment. As stated in the introduction, the effect of CSG on the function of proteins is various, and only a few computational studies have been led to describe the underlying molecular mechanisms. We added some more examples of CSG effect on protein function (more from experimental observations) in the introduction to highlight more efficiently the biological relevance of the post-translational modification. Besides and as above-mentioned, the S-glutathionylation does not impact the function of the proteins used as references in our work, which does not allow to further investigate binding free energies or molecular mechanisms behind functional changes. We hope that our CSG parameters set will be used in this sense in future computational studies.
Reviewer 2 Report
This is a quite good paper that can be recommonded to be published at IJMS. Developing force field for PTM is long -term and difficult research project, and this paper is foucing the field by studiing S-glutathionylation. The MS result also shoe the robust of these AMBER force field parameters. I think this paper is important, and thus need some more deail of how to used thisforce field for more users. I suggest to add a tutorial in the method part or supplementary, at least in the GitHub.
Author Response
This is a quite good paper that can be recommonded to be published at IJMS. Developing force field for PTM is long -term and difficult research project, and this paper is foucing the field by studiing S-glutathionylation. The MS result also shoe the robust of these AMBER force field parameters. I think this paper is important, and thus need some more deail of how to used thisforce field for more users. I suggest to add a tutorial in the method part or supplementary, at least in the GitHub.
We thank the reviewer for her/his useful suggestion, which will indeed help anyone who would use these parameters. Our parameters set is in the AMBER type, meaning that is is usable as any other parameter files when creating the initial topology and coordinates files for the S-glutathionylated system of interest. There are already very good tutorials available to learn how to do so on the AMBER website. We added a mention and a link to these tutorials on our Github page (in the README file), and also added a short sentence in the '2.2 System setup' section to clarify this point.
Reviewer 3 Report
Glutathionylation is a reversible posttranslational modification that protects protein cysteines from irreversible oxidation during oxidative stress. Glutathionylation modifies protein function, adapting its function to the state of oxidative stress in the cell. Robust AMBER force field parameters for S-glutathionylated cysteines proposed by the authors will allow the modelling of glutathionylated proteins by molecular dynamics, which is relevant. The article is well written, the data are presented clearly and logically, and it is well illustrated. Accept as is
Author Response
"Glutathionylation is a reversible posttranslational modification that protects protein cysteines from irreversible oxidation during oxidative stress. Glutathionylation modifies protein function, adapting its function to the state of oxidative stress in the cell. Robust AMBER force field parameters for S-glutathionylated cysteines proposed by the authors will allow the modelling of glutathionylated proteins by molecular dynamics, which is relevant. The article is well written, the data are presented clearly and logically, and it is well illustrated. Accept as is"
We thank the reviewer for carefully reading our manuscript and for her/his positive comments. We are glad he/she appreciated the reading of our manuscript and found it sound enough for publication as is.
Reviewer 4 Report
The authors developed force-field parameters for studying S-glutathionylated cysteines in proteins and evaluated the ability of the parameters to describe the global and local structures of eleven proteins with microsecond molecular dynamics simulation. Besides using crystal structures to start the molecular dynamics simulation, they also used structures predicted by AlphaFold. The simulation starting with the crystal structures showed good agreements with the experimental structures. Simulations starting from the structures predicted by AlphaFold worked reasonably well for most systems although not as well for a few. The disagreement appeared to result from local structures not predicted probably from AlphaFold and regular molecular dynamics simulations were not capable of fixing the structures within microsecond of simulation time. Overall, the study showed that the force field parameters developed performed reasonably well in describing structural properties. Scientists should find these parameters useful in simulating proteins containing S-glutathionylated cysteines.
The authors might want to fix missing or extra words in some sentences.
The authors might want to fix missing or extra words in some sentences.
Some word choices could be improved.
However, readers should have no problem understanding the science presented.
Author Response
The authors developed force-field parameters for studying S-glutathionylated cysteines in proteins and evaluated the ability of the parameters to describe the global and local structures of eleven proteins with microsecond molecular dynamics simulation. Besides using crystal structures to start the molecular dynamics simulation, they also used structures predicted by AlphaFold. The simulation starting with the crystal structures showed good agreements with the experimental structures. Simulations starting from the structures predicted by AlphaFold worked reasonably well for most systems although not as well for a few. The disagreement appeared to result from local structures not predicted probably from AlphaFold and regular molecular dynamics simulations were not capable of fixing the structures within microsecond of simulation time. Overall, the study showed that the force field parameters developed performed reasonably well in describing structural properties. Scientists should find these parameters useful in simulating proteins containing S-glutathionylated cysteines.
The authors might want to fix missing or extra words in some sentences
Some word choices could be improved.
However, readers should have no problem understanding the science presented.
We are grateful for the positive reviewer’s comment about our work. We revised our manuscript to improve the English language and to remove mistypes, and we hope it will now be even clearer to the readership.